# Torrent Poisoning Protection with a Reverse Proxy Server

António Godinho [1], José Rosado [1,2], Filipe Sá [1,3], Filipe Caldeira [3,4,*] and Filipe Cardoso [2,4]

1   Coimbra Polytechnic, Coimbra Institute of Engineering, Rua Pedro Nunes, Quinta da Nora,
    3030-199 Coimbra, Portugal
2   INESC Coimbra—Instituto de Engenharia de Sistemas e Computadores de Coimbra, Rua Sílvio Lima, Pólo II,
    3030-790 Coimbra, Portugal
3   CISeD—Research Centre in Digital Services, 3504-510  Viseu, Portugal
4   Viseu Polytechnic, School of Technology and Management of Viseu, Av. Cidade Politécnica,
    3504-510 Viseu, Portugal
*   Correspondence: caldeira@estgv.ipv.pt; Tel.: +351-232-480-500

**Abstract:** A Distributed Denial-of-Service attack uses multiple sources operating in concert to attack a network or site. A typical DDoS flood attack on a website targets a web server with multiple valid requests, exhausting the server's resources. The participants in this attack are usually compromised/infected computers controlled by the attackers. There are several variations of this kind of attack, and torrent index poisoning is one. A Distributed Denial-of-Service (DDoS) attack using torrent poisoning, more specifically using index poisoning, is one of the most effective and disruptive types of attacks. These web flooding attacks originate from BitTorrent-based file-sharing communities, where the participants using the BitTorrent applications cannot detect their involvement. The antivirus and other tools cannot detect the altered torrent file, making the BitTorrent client target the webserver. The use of reverse proxy servers can block this type of request from reaching the web server, preventing the severity and impact on the service of the DDoS. In this paper, we analyze a torrent index poisoning DDoS to a higher education institution, the impact on the network systems and servers, and the mitigation measures implemented.

**Keywords:** torrent poisoning; index poisoning; HAProxy; reversed proxy; Distributed Denial-of-Service (DDoS) flooding attack





## 1. Introduction

When successful, Denial-of-Service (DoS) attacks may stop legitimate users from accessing a specific network resource such as a web server [1]. Distributed DoS (DDoS) attack [2] is more effective because the attack originates from multiple sources, is more difficult to block, and is more effective in data load. Index poisoning of a torrent file is a type of DDoS attack that uses the BitTorrent protocol. Peer-to-peer file-sharing systems are still the most popular method for any type of content sharing over the internet, especially using the BitTorrent protocol. Like any other technology using the internet, it is vulnerable to attacks, suffering from different kinds of denial-of-service attacks, such as query flooding and pollution [3]. For example, in the content pollution attack, a malicious user publishes a large number of decoys (same or similar meta-data) so that queries of a given content return predominantly fake/corrupted copies [4]. This type of attack has been used by media corporations to fight back against their copyrighted material being shared, inserting corrupted files, and making those copies useless. Examples include HBO with the TV show Rome in 2005 and MediaDefender with the movie Sicko in 2007. This work analyses the impact of a flooding attack using torrent index poisoning and aims to enlighten and provide helpful information on mitigating a DDoS attack. It explores reverse proxy features, which provide excellent protection to mitigate these attacks when adequately applied and tuned. The document is organized as follows: After this Introduction, the peer-to-peer

and BitTorrent protocols are described in Section 2. Some types of attacks are presented in Section 3. Next, in Section 4, the case study is described and the kind of measures that were used to mitigate the attack. The results are presented in Section 5. The Discussion is in Section 6. Finally, Section 7 gives the Conclusions.

## 2. Peer-to-Peer and BitTorrent

BitTorrent is a P2P protocol developed on a distributed peer-to-peer network file-sharing system. BitTorrent is one of the most common protocols for transferring large files, with many Linux distributions including a torrent link for downloading their ISO files. To send or receive files, the user must have a BitTorrent client, a program that implements the BitTorrent protocol [5].

*BitTorrent Protocol*

The BitTorrent protocol uses metadata files called ***.torrent*** files or a specific type of hyperlink called magnet links. These BitTorrent files describe the content to be shared, and the announcers are the tracker servers. Tracking or indexers are servers that keep a list of IP addresses of all users/clients that are downloading a torrent (Figure 1). The list is sent to any new client connecting (called a peer), which will connect to a swarm to download/share the file content. A torrent file usually consists of a filename with the ***.torrent*** extension that contains the list of various torrent trackers to obtain the peers lists. The clients with that list can connect directly to each other using peer-to-peer (P2P). All clients contact the trackers regularly to receive an updated list of their peers.

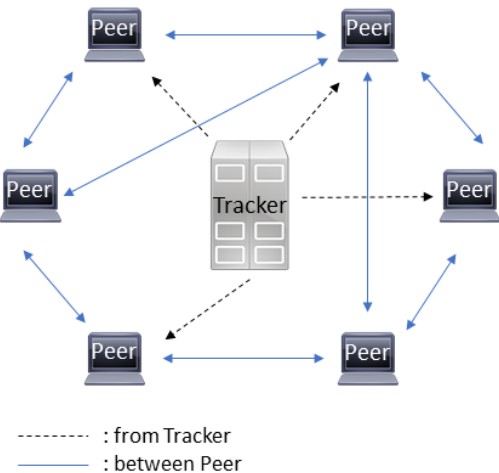

**Figure 1.** BitTorrent Infrastructure.

The magnet links are hyperlinks that give direct access to the torrent hash file, allowing the user to download the correspondent torrent file. These magnet links contain a unique identifier, which is a cryptographic hash of the torrent files. Using this link, the client will join a swarm using a separate peer-to-peer network that works with a Distributed Hash Table (DHT). Then, the client will announce that it is downloading/sharing the contents with a magnet link hash, asking for peers. The DHT network contains several nodes that act like a tracker server that the clients use to retrieve the list of peers. These types of links can include a link to a tracker in the case of the failure of the DHT network.

## 3. Types of Attacks
### 3.1. DoS and DDoS

The are several types of DDoS flood attacks. The most common types try to disrupt the service by

- Exhausting bandwidth–router processing capacity or network resources; or
- Exhausting the server resources—CPU, memory, disk/database bandwidth, and I/O bandwidth [6].

This work focuses on the web servers flooding attack (Figure 2). It is very important to understand that from the server perspective, the incoming requests are legitimate, and the server will process them as if a normal user was accessing a web resource. This is the main reason why flooding the target server with requests is so effective. The server will try to process all requests until it cannot process and respond to them or just crashes, preventing the service from being provided to legitimate users. The attack described in this paper falls under the type of attack that exhausts the server resources. A Distributed Denial of Service (DDoS) [7] is a coordinated DoS from multiple sources, significantly increasing the number of requests reaching the server. With that, the attack efficiency increases, making this type of attack more difficult to block, since there are multiple sources, and usually, new attackers keep joining the attacking group. Furthermore, it is harder to truly identify the attacker origin in this kind of attack.

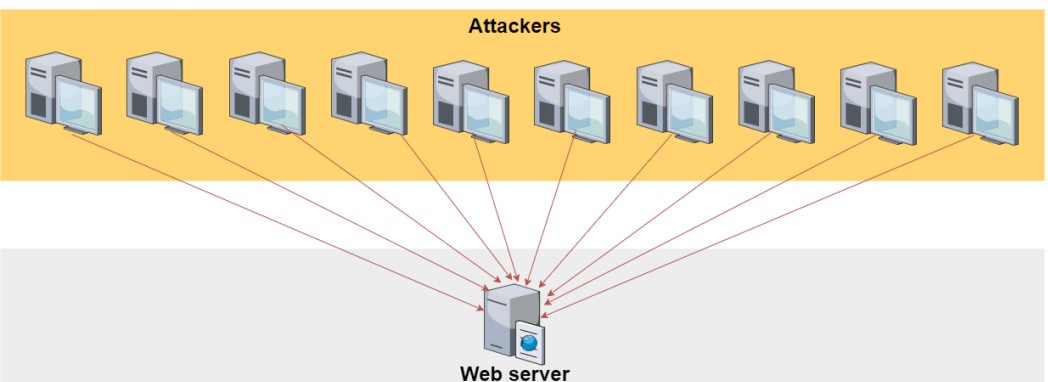

**Figure 2.** DDos attack.

### 3.2. Flooding Attack

A Distributed Denial-of-Service (DDoS) flooding attack may be classified into two main categories: network/transport-level DDoS flooding attacks and application-level DDoS flooding attacks. Application-level DDoS attacks generally consume less bandwidth and are stealthier than volumetric attacks since they are very similar to valid traffic. However, application-level DDoS flooding attacks usually have the same impact on the services since they target specific characteristics of applications such as HTTP [2]. There are several subtypes of application-level DDoS attacks that use the HTTP/S protocol, more specifically HTTP flooding attacks: session flooding attacks, request flooding attacks, asymmetric attacks, and slow request/response attacks [2].

Under continued attack-related congestion, flow-controlled applications will continue to increase their back-off time between re-transmissions. From the user's perspective, their workload is not being processed; a DoS situation has occurred [8].

### 3.3. Torrent Poisoning Attack

The torrent poisoning attack is the sharing of corrupt, virus-infected, or misleading file names using the BitTorrent protocol. The "copyright industry" uses pollution attacks to corrupt the targeted content and share it, rendering it unusable [9]. Unable to distinguish polluted files from unpolluted files, unsuspecting users download the contaminated files into their file-sharing folders, from which other users may then later download the polluted files. In this manner, polluted files spread through the file-sharing system. Often, users look for torrents with a greater number of peers, increasing the chances of a successful download. An example of this type of attack is one that occurred on 1 June 2022, in which a Google Cloud Armor customer was targeted with a series of HTTPS DDoS attacks, which peaked at 46 million requests per second. This is the largest Layer 7 DDoS reported to date—at

least 76% larger than the previously reported record. Approximately 22% (1169) of the source IPs corresponded to BitTorrent exit nodes, although the request volume from those nodes represented just 3% of the attack traffic. They believe that BitTorrent's participation in the attack was incidental due to the nature of the vulnerable services, even at 3% of the peak (more than 1.3 million requests per second). Their analysis shows that BitTorrent exit nodes can send a significant amount of unwelcome traffic to the web applications and services [10].

### 3.4. Torrent Index Poisoning Attack

Trackers are fundamental for the BitTorrent protocol since they are the servers that share the peers list, being contacted regularly by the clients. A torrent file contains several pieces of information about the file(s) to download, trackers, hash, etc. For example, from the Debian web site, it is possible to download the Linux distribution via torrent. To analyze the latest version, the .torrent provides the information in Figure 3.

```json
1 ▾ {
2       "announce": "http://bttracker.debian.org:6969/announce",
3       "comment": "\"Debian CD from cdimage.debian.org\"",
4       "created by": "mktorrent 1.1",
5       "creation date": 1662813552,
6 ▾     "info": {
7           "length": 400556032,
8           "name": "debian-11.5.0-amd64-netinst.iso",
9           "piece length": 262144,
10          "pieces": "<hex>C4 C5 62 14 1D 86 09 5B E2 F4 07 98 75 7F 9D B6 5D 12 8E 6D 12 66 48 DF 48 F3 48
11      },
12 ▾    "url-list": [
13          "https://cdimage.debian.org/cdimage/release/11.5.0/amd64/iso-cd/debian-11.5.0-amd64-netinst.iso",
14          "https://cdimage.debian.org/cdimage/archive/11.5.0/amd64/iso-cd/debian-11.5.0-amd64-netinst.iso"
15      ]
16 }
```

**Figure 3.** Debian 11.5.0 torrent file info.

The first step in this kind of attack is to alter a torrent file and insert the target IP address and port number into the tracker list, and then the clients will flood the victims' IP address and service running on that port. Figure 3 also clarifies how clients connect to trackers, since the announcer links are simple HTTP/S URLs. Although it is common to use port 6969, these servers may use the standard HTTP and HTTPS ports, TCP 80 and 443. Using the torrent file with a victim IP address, BitTorrent clients will not distinguish the victim, which they believe is a tracker. The target server will reply to the HTTP/S request with an invalid answer, which is how the client will understand it, but since the server is alive and replying, the clients keep retrying. These changed torrent files by themselves may render the attack unsuccessful. The final step is to attract the maximum number of users to download and use this file, publishing this file on torrent websites. Some of the websites where the .torrent file is published and shared require a minimum number of seeders (users sharing the full files content of the torrent) and leechers. Using a modified tracker, the attacker fakes the number of peers (leechers and seeders) connected and provides a high number of these participants. These numbers can deceive those websites with fake high statistics to the modified .torrent files. Furthermore, to be noted, users tend to download files with a high number of peers, making these torrents more appealing [10].

## 4. Case Study—Attack on a Higher Education Institution

The institution in the case study has over 3500 users, between students and staff. Due to the SARS-CoV-2 pandemic, an e-learning Moodle cluster solution was implemented [11]. With servers capable of responding to the requirements of distance learning, a system based on the premises of High Availability, High Performance, Load Balancing was implemented, as shown in Figure 4. Many of the institution's teachers had almost one and a half years of experience implementing and using Moodle for online exams and

student evaluation. During this time, the system proved to be robust, even on exams with a high number of students, several times with over 500 students simultaneously. On the afternoon of the 8th of September of 2021, while students had an exam, an attack hit the network.

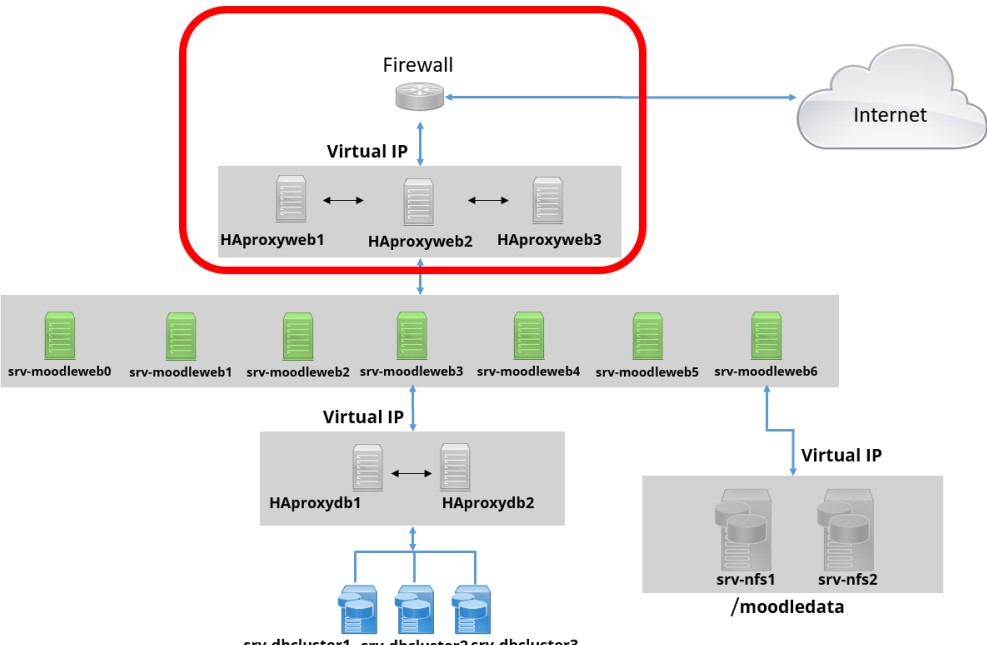

**Figure 4.** Moodle cluster Infrastructure.

In the middle of the afternoon, the network was unresponsive, all connections to the outside were extremely slow or timed out, and the Moodle cluster had the same behavior. Two main points were severely affected by the way the cluster was designed. First, the perimeter firewalls could not process the high volume of incoming requests, becoming unresponsive via the management console. Second, the reverse proxy cluster was also unable to respond to all requests (shown in the red area in Figure 4).

*4.1. Perimeter Firewalls*

The firewall cluster protecting the outside border of the institution was composed of a cluster of two firewalls configured in high availability. Regular daily traffic is usually around 30 to 50 Mbps, inbound and outbound. As shown in Figure 5, during the attack, the incoming traffic was as high as 800 Mbps for several hours, which is 15 times above the normal traffic of the institution.

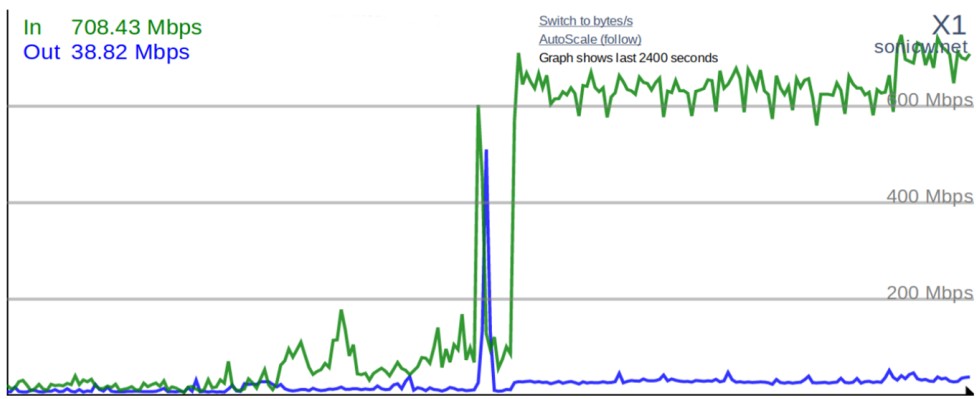

**Figure 5.** Firewall traffic.

### 4.2. Reverse Proxy Cluster

All incoming traffic to the web servers passes through the firewalls and is directed to the cluster of the reverse proxy servers running HAProxy. The cluster is composed of thrual servers running on a VMware ESXI cluster in high availability. Each of the reserve proxy servers is configured with a different level of priority, making one the master, the other the first backup, and the last one the second backup. Each node uses the Keepalived software, which uses the Virtual Router Redundancy Protocol (VRRP), sending multicast messages using the 112 protocol [11]. The Keepalived checks every two seconds if HAProxy is running and messages the other nodes.

The high number of requests hitting the master resulted in warnings on the ESXI, for CPU and memory usage (Figure 6), even in servers with 16 cores and 16 GB of RAM. The master never changed to one of the backups because the server and proxy service were up and running and were just unable to provide the service.

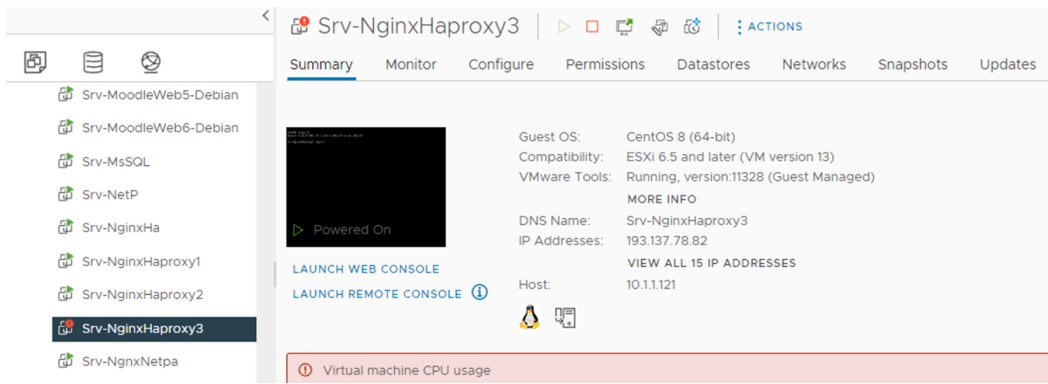

**Figure 6.** ESXI CPU usage warning.

While accessing the reverse proxy server logs for analysis, three things were clear. First, the huge quantity of requests sent to the servers made it impossible to read with a tail command due to the speed at which the text scrolled. Second, the kind of requests that were being sent to the server all had the path "/announce/" with a query string info_hash in the URL, as shown in Figure 7 with yellow boxes. Third, the browser that was accessing the URL did not show an expected name, such as Firefox, Chrome, Edge, and so on, but the BitTorrent client uTorrent, as shown in Figure 7, with red boxes.

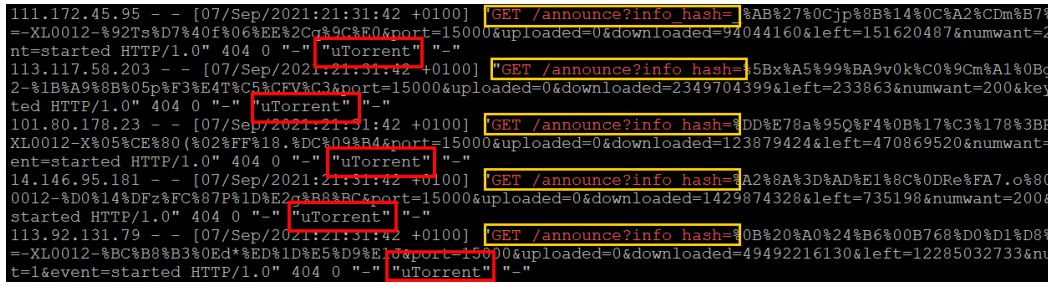

**Figure 7.** Web Log's relative URL/announce?info_hash=.

This attack continued for several days. The reserve proxy logs were kept to generate charts using the GoAcess-webstat tool. The regular size of the compressed log file each day was between 300 and 500 Kb (shown in Figure 8 in yellow box). The huge number of requests received during the attack resulted in files of over 300 megabytes. For stats, the institution uses the GoAcess-webstat tool, which is configured to automatically generate stats for all websites. As the log files increased (which was caused by the attack), this was unable to process files, and the application crashed.

```
[root@srv-webstats goaccess-processed]# ls -la wwwisec-ssl-access.log-202109* --block-size=K
-rw-r--r--. 1 nfsnobody nfsnobody    502K Dec 19 10:28 wwwisec-ssl-access.log-20210901.gz
-rw-r--r--. 1 nfsnobody nfsnobody    525K Dec 19 10:29 wwwisec-ssl-access.log-20210902.gz
-rw-r--r--. 1 nfsnobody nfsnobody    507K Dec 19 10:29 wwwisec-ssl-access.log-20210903.gz
-rw-r--r--. 1 nfsnobody nfsnobody    474K Sep  5  2021 wwwisec-ssl-access.log-20210904.gz
-rw-r--r--. 1 nfsnobody nfsnobody    280K Sep  6  2021 wwwisec-ssl-access.log-20210905.gz
-rw-r--r--. 1 nfsnobody nfsnobody    325K Sep  7  2021 wwwisec-ssl-access.log-20210906.gz
-rw-r--r--. 1 nfsnobody nfsnobody   2847K Dec 19 10:25 wwwisec-ssl-access.log-20210907.gz
-rw-r--r--. 1 nfsnobody nfsnobody 175842K Dec 19 10:25 wwwisec-ssl-access.log-20210908.gz
-rw-r--r--. 1 nfsnobody nfsnobody 356632K Dec 19 10:25 wwwisec-ssl-access.log-20210909.gz
-rw-r--r--. 1 nfsnobody nfsnobody 342602K Dec 19 10:25 wwwisec-ssl-access.log-20210910.gz
-rw-r--r--. 1 nfsnobody nfsnobody  31499K Dec 19 10:25 wwwisec-ssl-access.log-20210911.gz
-rw-r--r--. 1 nfsnobody nfsnobody  14208K Dec 19 10:25 wwwisec-ssl-access.log-20210912.gz
-rw-r--r--. 1 nfsnobody nfsnobody  11527K Dec 19 10:25 wwwisec-ssl-access.log-20210913.gz
-rw-r--r--. 1 nfsnobody nfsnobody   9039K Dec 19 10:25 wwwisec-ssl-access.log-20210914.gz
-rw-r--r--. 1 nfsnobody nfsnobody   7617K Dec 19 10:25 wwwisec-ssl-access.log-20210915.gz
```

**Figure 8.** Log file size in September.

A deeper analysis of the log file size for this work provided other unknown information at the time. The attack happened again in October and occurred in two different waves. In the list of the ten biggest log files of all time for the institution website (Figure 9), all are from October; eight files were between 700 and 950 megabytes, another was near 1.5 gigabytes, and the last one was over 4 gigabytes but uncompressed. For the last log file, it was uncompressed because the log rotation process (Linux logrotated daemon) was not able to compress the existing file on rotation due to the lack of disk space. This was another consequence of this attack, resulting in the reverse proxy servers disk becoming full of web logs.

```
[root@srv-webstats goaccess-processed]# ls -la * --block-size=M --sort=size -l | head -10 | grep wwwise
-rw-r--r--. 1 nfsnobody nfsnobody 4397M Oct 27  2021 wwwisec-ssl-access.log-20211026
-rw-r--r--. 1 nfsnobody nfsnobody 1575M Oct 28  2021 wwwisec-ssl-access.log-20211027.gz
-rw-r--r--. 1 nfsnobody nfsnobody  948M Oct 29  2021 wwwisec-ssl-access.log-20211028.gz
-rw-r--r--. 1 nfsnobody nfsnobody  932M Nov  1  2021 wwwisec-ssl-access.log-20211031.gz
-rw-r--r--. 1 nfsnobody nfsnobody  881M Oct 30  2021 wwwisec-ssl-access.log-20211029.gz
-rw-r--r--. 1 nfsnobody nfsnobody  828M Nov  2  2021 wwwisec-ssl-access.log-20211101.gz
-rw-r--r--. 1 nfsnobody nfsnobody  821M Oct 31  2021 wwwisec-ssl-access.log-20211030.gz
-rw-r--r--. 1 nfsnobody nfsnobody  749M Nov  3  2021 wwwisec-ssl-access.log-20211102.gz
-rw-r--r--. 1 nfsnobody nfsnobody  744M Nov 15  2021 wwwisec-ssl-access.log-20211114.gz
-rw-r--r--. 1 nfsnobody nfsnobody  743M Nov 28  2021 wwwisec-ssl-access.log-20211127.gz
[root@srv-webstats goaccess-processed]#
```

**Figure 9.** Ten of the biggest log files of all time.

### 4.3. Accepting and Validating the Client Requests

This attack type implies using the target IP address or server name. The IP address is often added to a rogue or malicious DNS server to disguise the target on the torrent file. In this case, for example, the IPv4 address of the institutional website was used, so the attackers could add this IP or use a valid one, but with a fake server address, for example, server100.trackerlist.something, which could be resolved to the same IP address. Then, the core problem is that the server should only accept requests to valid websites that this server is running [12].

#### SNI—Server Name Indication

Transport Layer Security (TLS) is a cryptographic protocol designed to provide communications security over a computer network, implementing an extension called Server Name Indication (SNI) [13]. Nowadays, one web server may provide multiple websites with only one IP address. With multiple websites, each with its own SSL certificate, the server must provide the correct certificate to the client. SNI is an extension of the TLS/SSL protocol, which is used on the HTTPS protocol [14]. It is included in the TLS/SSL handshake process in order to ensure that client devices are able to see the correct SSL certificate for the website they are trying to access. With this extension, the client specifies the hostname or domain name of the website during the TLS handshake instead of when the HTTP connection opens after the handshake negotiation. With that, this verification is completed at the very beginning of the connection.

### 4.4. HAProxy

HAProxy [15], which stands for High Availability Proxy, is a popular open-source software TCP/HTTP Load Balancer and proxying solution. Its most common use is to improve the performance and reliability of a server environment by distributing the workload across multiple servers (web servers, applications, and databases). It is used in many high-profile environments, including GitHub, Imgur, Instagram, and Twitter [16]. The configuration of HaProxy is based on defining the interfaces that receive requests (frontends) and those that connect to internal servers (backends), implying the definition of at least a frontend and a backend. Since version 1.8, HAProxy has supported SNI and SNI health checks. Since HAProxy is a reverse proxy server, it can work in HTTP or TCP proxy, and both may be used with SNI [17].

#### 4.4.1. HAProxy Mode HTTP

The most usual scenario implemented with HAProxy is shown in Figure 10, acting as a reverse proxy server. The clients connect to the HAProxy, which decides if a connection to an internal server is made. The clients can only see the perimeter server, which will act as a web server, as shown in Figure 10. HAProxy operates at layer 7 with the backend servers using this format. Operating in HTTP mode, HAProxy can extract the SNI (ssl_fc_sni) from the request or the host header information (hdr(host)) to filter the incoming session and route it to the proper backend server.

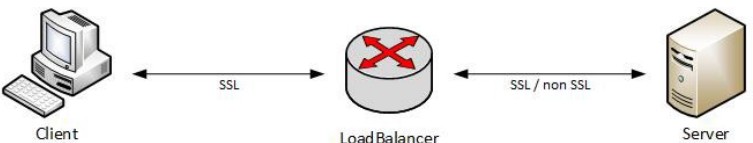

**Figure 10.** Mode HTTP—SSL Termination.

#### 4.4.2. HAProxy Mode TCP

When configured in TCP mode (Figure 11), HAProxy will operate at layer 4, the TCP layer. In this case, traffic passes to the backends via HAProxy, and the HTTPS data are not decrypted, so the reverse proxy server cannot analyze what is being exchanged between the client and server. Still, at the beginning of the connection, on the TLS/SSL ClientHello, HAProxy can extract SNI from the first client hello, before the TLS session is even established. This way, it can filter the session using the req_ssl_sni value. Neither the ssl_fc_sni indicator nor the host header information (hdr(host)) is a valid option in TCP mode [18].

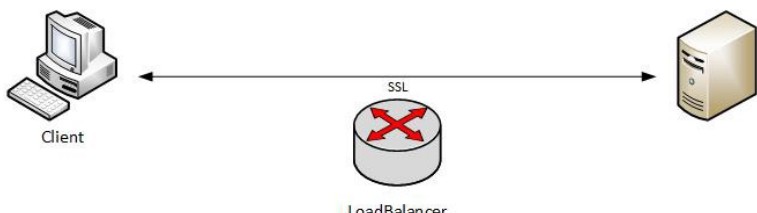

**Figure 11.** Mode TCP—Transparent passthrough.

#### 4.4.3. HAProxy Configuration

In HAProxy, ssl_fc_sni and req_ssl_sni are used with Access Control Lists (ACLs). The objective is to only accept traffic to a server with a valid name, for example, the institutional website address name. The configuration file for HProxy contains several sections. When HAProxy is configured as a reverse proxy, it has to define two sections: frontend and backend. Since an instance of HAProxy may have multiple frontends and backends, the frontend section has a set of rules that define what client requests should be accepted and how to forward them to the backends. The backend section defines the internal server

or servers that will receive forwarded requests. Although HAProxy is configured in HTTP mode, a small portion of the configuration is related to TCP. The HAProxy documentation page states that content switching is needed, and it is recommended to first wait for a complete client hello (type 1) [19].

Listing 1 is a portion of the front and backend configuration used to mitigate the attack. If the SNI matches that name, HAProxy will proxy the request to the backend server. If the SNI does not match, it is assumed that it is an invalid/illegal access, and the request should be discarded. In this case, HAProxy will make a TCP request reject. This option will close the connection without a response once a session has been created but before the HTTP parser has been initialized.

**Listing 1.** HAProxy configuration for the institutional address website (relevant parts only).

```
 1  frontend                 www_frontend
 2  mode http
 3  bind *:80
 4  bind *:443 ssl crt /etc/haproxy/ssl/institutional_address_bundle.
        pem alpn h2,http/1.1
 5  tcp-request inspect-delay 5s
 6  tcp-request content accept if { req_ssl_hello_type 1 }
 7  http-request redirect scheme https unless { ssl_fc }
 8  http-response set-header Strict-Transport-Security max-age
        =31536000
 9  acl backend_1 req_ssl_sni -i institutional_address.website
10  acl backend_1 ssl_fc_sni -i institutional_address.website
11  acl backend_1 req_ssl_sni -i institutional_address.domain_name
12  acl backend_1 ssl_fc_sni -i institutional_address.domain_name
13  use_backend www_backend if backend_1
14  default_backend~bk_drop
15
16  backend                  www_backend
17  ... valid requests through here
18  backend bk_drop
19  mode http
20  tcp-request content reject
```

## 5. Results

The HAProxy Stats page is a valuable resource for real-time information. However, this page was unresponsive and impossible to access during the attack. Another resource is GoAcess-webstat, a tool to generate stats that could be useful for understanding the impact on the firewall and the servers. Unfortunately, the size of the logs made the application useless since it could not process the amount of information recorded on the logs files. The tool has another limitation, which is that it is only able to show the stats from the last 12 months.

### 5.1. Hits and Requests—Year 2020

When studying the website traffic behavior during the year 2020, some observations were found that we must observe carefully. These occurred during the exams period at the end of each semester (February and June) and September and at the beginning of October, when students verified their classes schedule. There were four other access peeks during the year, but none over the ones on this period.

On 6 October, there were 7914 visitors, while the year average was 1590. On 2 October, there were 126,232 hits, while the average was 28,748 (Figure 12). Using this tool, there are other indicators to point out:

- The most requested URL/File was /PT/Default.aspx with over 546k hits (4.57% of all accesses).

- The total sum of 404 requests was around 562k hits, with 380 MB in responses.

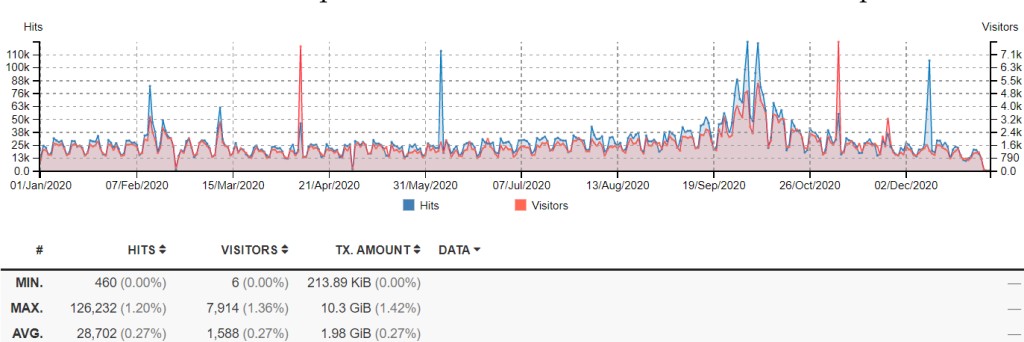

| # | HITS ⇕ | VISITORS ⇕ | TX. AMOUNT ⇕ | DATA ▾ | |
|---|---|---|---|---|---|
| MIN. | 460 (0.00%) | 6 (0.00%) | 213.89 KiB (0.00%) | | — |
| MAX. | 126,232 (1.20%) | 7,914 (1.36%) | 10.3 GiB (1.42%) | | — |
| AVG. | 28,702 (0.27%) | 1,588 (0.27%) | 1.98 GiB (0.27%) | | — |

**Figure 12.** Institutional website—2020 access stats.

In the context of this type of attack, the impact of 404 status codes [20] is extremely important to understand.

In the year 2020, there were a total of 1,021,899 requests that resulted in a 404 error, with a percentage of 8.54% of all requests of that year, as shown in Figure 13.

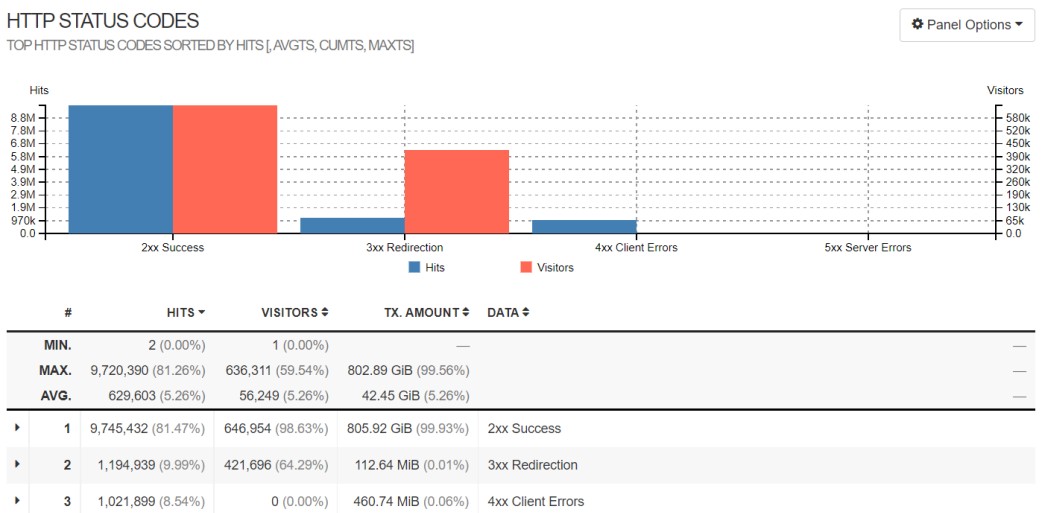

| # | HITS ▾ | VISITORS ⇕ | TX. AMOUNT ⇕ | DATA ⇕ | |
|---|---|---|---|---|---|
| MIN. | 2 (0.00%) | 1 (0.00%) | — | | — |
| MAX. | 9,720,390 (81.26%) | 636,311 (59.54%) | 802.89 GiB (99.56%) | | — |
| AVG. | 629,603 (5.26%) | 56,249 (5.26%) | 42.45 GiB (5.26%) | | — |
| ▸ 1 | 9,745,432 (81.47%) | 646,954 (98.63%) | 805.92 GiB (99.93%) | 2xx Success | |
| ▸ 2 | 1,194,939 (9.99%) | 421,696 (64.29%) | 112.64 MiB (0.01%) | 3xx Redirection | |
| ▸ 3 | 1,021,899 (8.54%) | 0 (0.00%) | 460.74 MiB (0.06%) | 4xx Client Errors | |

**Figure 13.** 2020 Top http status codes sorted by hits.

## 5.2. Hits and Requests Until August 2021

On 6 January, there were 20,275 visitors, whereas the average that year was 1559. Previously during 2020, it were 1590. The 22 and 23 of January were record days in the number of hits, with 142,357 and 203,408, respectively, while the average was 27,913 and previously 28,748 (Figure 14).

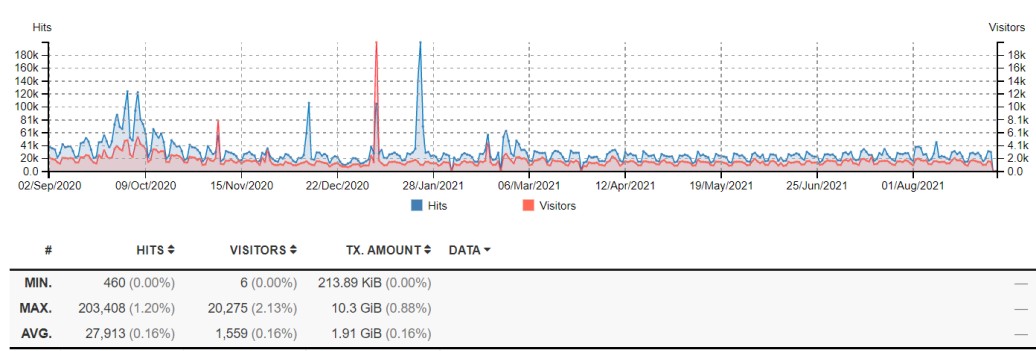

| # | HITS ⇕ | VISITORS ⇕ | TX. AMOUNT ⇕ | DATA ▾ | |
|---|---|---|---|---|---|
| MIN. | 460 (0.00%) | 6 (0.00%) | 213.89 KiB (0.00%) | | — |
| MAX. | 203,408 (1.20%) | 20,275 (2.13%) | 10.3 GiB (0.88%) | | — |
| AVG. | 27,913 (0.16%) | 1,559 (0.16%) | 1.91 GiB (0.16%) | | — |

**Figure 14.** Institutional website—before September 2021.

The chart with the access until September 2021 is consistent with that of the year before. The results in 2020 are similar. The 404 requests also remain constant, with a percentage of 8.11% of all requests for that year, as shown in Figure 15.

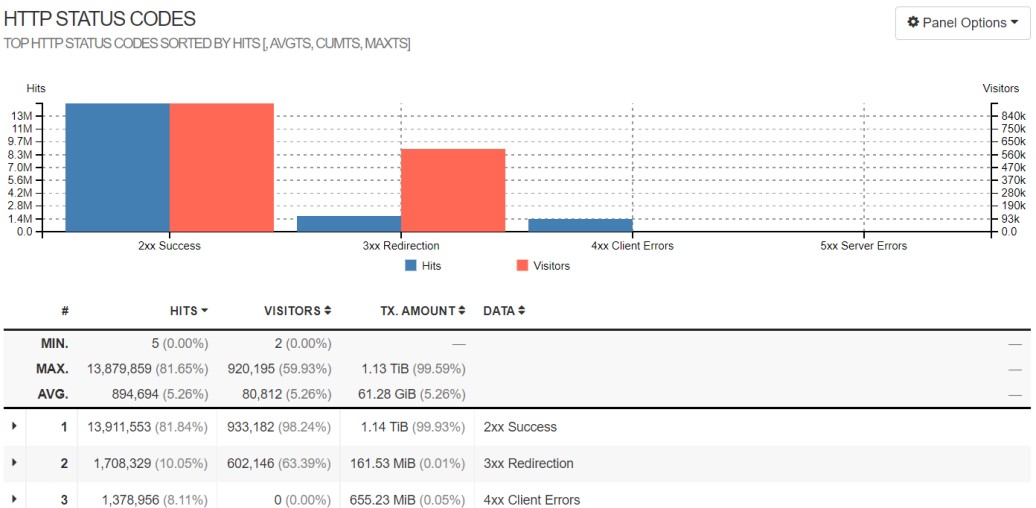

**Figure 15.** Top HTTP status codes in 2021 sorted by hits.

### 5.3. Hits and Requests—September 2021

With the attack, the record of hits requests to the website made the remaining 11 months irrelevant, while the max number of visitors stayed the same, meaning that were never over 21 thousand visitors per day. The number of hits explains the type of attack, and the limit on the axis on the left changes two hundred thousand to five million, as shown in Figure 16.

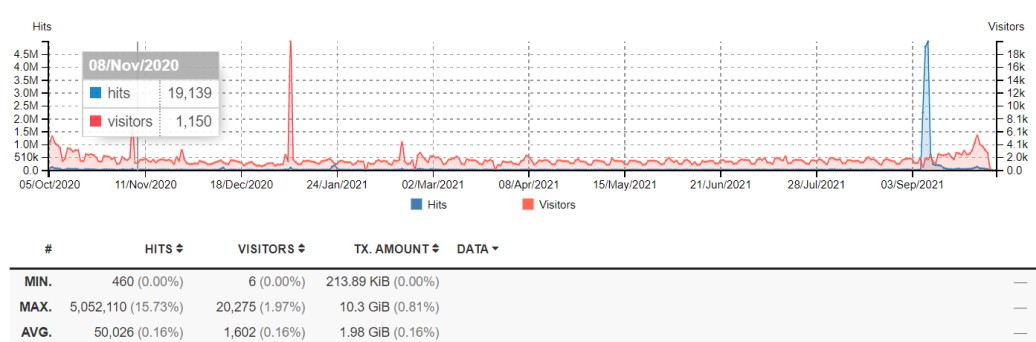

**Figure 16.** Institutional website hits—September 2021.

There were over 5 million hits on the 9th of September. The seven days with top hits were 9th, 8th, 7th, 10th, 11th, 12th, and 13th in descending order.

The 404 requests skyrocket to over fifteen million, with the percentage of that type of requests passing from 8.54% to 46.89%, as shown on Figure 17. The number of 404 requests is similar to the number of valid requests (2xx) of the last year, almost a 40% growth in twelve months.

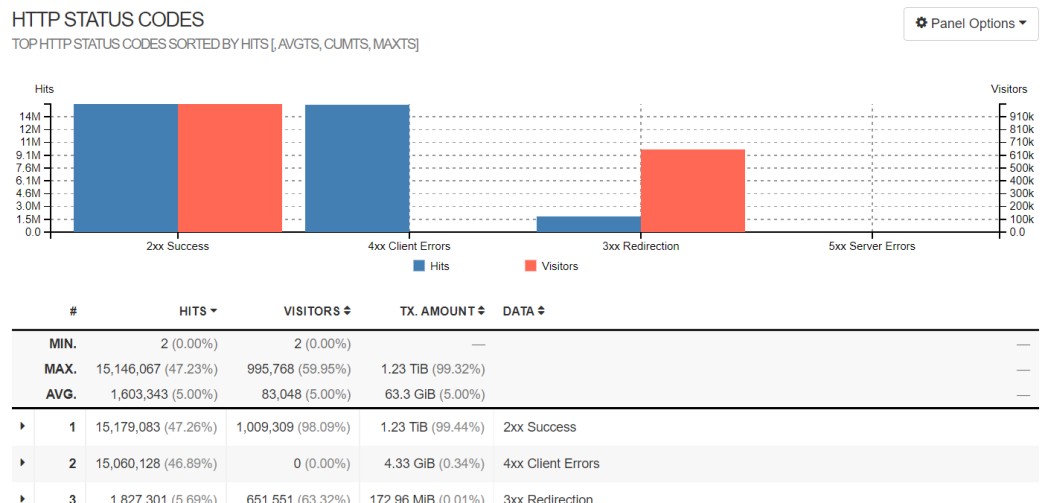

**Figure 17.** HTTP status codes for the institutional website—September 2021.

## 6. Discussion

Deep logging analysis is extremely helpful in understanding the nature of a DDoS attack. Without this knowledge, stopping, blocking, or at least minimizing the impact of this type of attack may be impossible. The Torrent Network design, especially for the websites that share torrent files and mag links, makes it extremely difficult to report and block files with invalid announcers. If this was possible, those kinds of attacks could be blocked at the source, but unfortunately, most of these websites share illegal or copyrighted software and movies. Using Open Source Reserve Proxy software is a reasonable solution for server protection. Other defense mechanisms using Open Source, such as the use of IPTABLES firewalls, can improve this protection by blocking incoming attackers, implementing a connection limit during a defined period of time, or analyzing the request type or content. In the Open Source software, there are also web application firewalls; some may be used as modules with reserve proxy solutions such as ModSecurity. The use of HAProxy configured with the Server Name Indication detection was the way found to minimize the impact of this kind of attack.

This work implements a solution using HAProxy, but other reserve proxy solutions may be used to obtain the same result. As demonstrated in Figure 4, the network was already designed using reserve proxy servers. Based on the same scenario, NGINX can be used as a reserve proxy server and block access to requests to the URL /announce.

NGINX is more than a reverse proxy solution. It is also a web server, meaning that the same configuration may be used as a web server (Listing 2). Other web servers such as Apache Web Server use a similar configuration and can also be used for the same end (Listing 3).

**Listing 2.** NGINX configuration block relative URL /announce (relevant parts only).

```
1  location /announce {
2  deny all;
3  return 404;
4  }
```

Even IIS, which is a different solution from the Linux alternatives, has a request filtering feature that allows denying access to specific URLs, returning an HTTP Error 404.5—URL Sequence denied error message.

By analyzing the stats charts, we can see that another essential element should be changed. The institution uses a template master page for its website. This template is used on all their website, including the error pages. This is a typical behavior that web developers use to create a single template web page, and they use it as a base for all others.

This master page includes the files required for almost all pages, including CSS styles files, JavaScript files, Font files, etc. These are a lot of useless elements for a static web page, such as an error 404 page. Using the master page for the error page means that at each attack request, the server would reply with all included files (CSS, JavaScript), lots of bandwidth, and server resources with useless information. This example shows that the error pages should be trimmed to the bare minimum, minimizing the impact of the attacks. Table 1 shows the size of the initial 404 web page and the current size. The same table is compared with the default 404 page from other reverse servers or browsers.

**Listing 3.** Apache configuration block relative URL /announce (relevant parts only).

```
1 <Location /announce>
2 Deny from all
3 </Location>
4 <FilesMatch announce*>
5 Deny from all
6 </FilesMatch>
```

**Table 1.** 404 page sizes.

| Old 404 Page | 404 Page | HAproxy 404 | NGINX 404 | IIS 404 |
|:---:|:---:|:---:|:---:|:---:|
| 34.42 KiB | 2.09 KiB | 2.08 KiB | 218 Bytes | 2.08 KiB |

The current 404 page was changed at the end of 2018 and did not have an impact on this attack. Taking into account the values on Table 1, still, the size of the webpage could have been optimized. There were over 13.5 million 404 requests in September. If the original version was still online, the total downloaded data from the web server was 443 Gib. Using the actual version, the total data amounted to 26.9 Gib, while using the default NGINX or HAProxy, the total was 2.74 GiB. That is a total 26.9 GiB difference from the actual version to an optimized version on a single attack.

On the server exposed to the internet, a firewall was configured. The server was running Debian Linux operating system with Uncomplicated Firewall (UFW), allowing only HTTP, HTTPS, and SSH. An analysis of the log files shows that the same source makes a new request at each second, as shown in Figure 18 with the red text color.

**Figure 18.** Requests by IP per time.

UFW uses iptables and may be configured to limit incoming network connections to a certain number per IP and port or limit concurrent connections. The institution had a cluster of firewalls protecting the perimeter, but there should also exist server protection. In this case, on the file /etc/ufw/before.rules, the configuration on Listing 4 should have been added.

**Listing 4.** Limiting number of connections per IP with ufw.

```
1  # Limit to 10 concurrent connections on port 443 per IP
2  -A ufw-before-input -p tcp --syn --dport 443 -m connlimit --
      connlimit-above 10 -j~DROP
3
4  # Limit to 20 connections on port 443 per 2 seconds per IP
5  -A ufw-before-input -p tcp --dport 443 -i eth0 -m state --state
      NEW -m recent --set
6  -A ufw-before-input -p tcp --dport 443 -i eth0 -m state --state
      NEW -m recent --update --seconds 2 --hitcount 20 -j DROP
```

The are also other firewall solutions for Linux distributions, such as Config Server Firewall (CSF) or Firewalld, and both allow similar configurations.

## 7. Conclusions

A configuration similar to the one presented in this work, applied on a reverse proxy server, can partially prevent an Index Torrent poisoning attack. After this implementation, the attack cannot reach the internal servers, only the perimeter firewall and reverse proxy. The connections are closed and discarded at the reverse proxy server, and no reply is sent back to the user. This way, the impact on these two elements is minimized, since no connection tracking from the outer firewall to the internal web servers was required. This type of tracking is one of the most demanding features in terms of memory and CPU for network devices and servers. Like many other DDoSs, all that systems administrators can do is try to minimize the impact of these types of attacks on their services. Still, in our case study the perimeter firewalls were almost unresponsive via their administration pages. Furthermore, any internal services open to the internet were practically unreachable, with lots of timeouts. The implemented solution uses only open-source software without any costs to the institution. The size of the error pages 4xx, and 5xx may influence the attack's impact. Using small optimized error web pages may result in a dozen gigabytes not being sent from the web servers to the attackers. Even with small changes, since one DDoS attack may result in millions of requests, the difference is significant. The lack of protection on the server firewall was an error. The most popular Linux distributions allow the installation of firewall tools—for Debian or Ubuntu-based, UFW or CSF. If one of these tools is used and a limitation to the number of connections per IP or concurrent connections is implemented, the attack could have been mitigated on the reserve proxy and web servers.

**Author Contributions:** Conceptualization, A.G., F.C. (Filipe Cardoso) and J.R.; software, A.G.; validation, F.S. and J.R.; formal analysis, F.S. and J.R.; investigation, A.G., F.C. (Filipe Cardoso) and F.S.; data curation, A.G. and F.S.; writing—original draft preparation, A.G.; writing—review and editing, F.C. (Filipe Caldeira), F.C. (Filipe Cardoso), F.S. and J.R.; supervision, F.C. (Filipe Cardoso); project administration, A.G., F.C. (Filipe Caldeira) and F.C. (Filipe Cardoso). All authors have read and agreed to the published version of the manuscript.

**Funding:** This work is partially funded by National Funds through the FCT—Foundation for Science and Technology, I.P., within the scope of the projects UIDB/00308/2020, UIDB/05583/2020 and MANaGER (POCI-01-0145-FEDER-028040). Furthermore, we would like to thank the Research Centre in Digital Services (CISeD) and the Polytechnics of Viseu and Coimbra for their support.

**Data Availability Statement:** Data are contained within the article.

**Conflicts of Interest:** The authors declare no conflicts of interest.

**Abbreviations**

The following abbreviations are used in this manuscript:

| | |
|---|---|
| ACK | Acknowledgment |
| ACL | Access Control Lists |
| CPU | Central Processing Unit |
| CSS | Cascading Style Sheets |
| DDoS | Distributed Denial of Service |
| DNS | Domain Name System |
| DoS | Denial-of-service |
| HTTP | Hypertext Transfer Protocol |
| HTTPS | Hypertext Transfer Protocol Secure |
| ICMP | Internet Control Message Protocol |
| IP | Internet Protocol |
| Mbps | Megabits per second |
| P2P | Peer-to-peer |
| RAM | Random Access Memory |
| SNI | Server Name Indication |
| SSL | Secure Sockets Layer |
| TCP | Transmission Control Protocol |
| TLS | Transport Layer Security |
| UDP | User Datagram Protocol |
| URL | Uniform Resource Locator |

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
