# Peer review of "Torrent Poisoning Protection with a Reverse Proxy Server"

_electronics, doi:10.3390/electronics12010165_

Round 1

Reviewer 1 Report

The paper shows a configuration method to be applied on a network to mitigate an index torrent poising attack. The idea is good; the authors use an interesting case study to demonstrate their work. However, I found it quite challenging to understand the paper's main contribution. The paper presentation needs much improvement. The abstract needs to be more precise and describe the main research problem the paper wants to address. In addition, the introduction section is relatively short, there needs to be a clear description of what the paper wants to say, and there are no related works in that context. Many small sections could be gathered into one or two to show a more dedicated view of the paper's research activities. There is no future vision regarding the paper's work.  

It would be great if the authors described the main research problem and how the paper‘s contribution is being used to address this problem more clearly.

Other minor points:

- Select more specific keywords for this work. 

- Try to use all white spaces in the paper, such as page 9. 

- Figures from 11 to 16 are essential, where the texts are difficult to be readable. 

Reviewer 2 Report

It is a good idea to use virtual infrastructure. The outcome is a technological solution, but it lacks scientific value. The result is not comparable with other existing scientific methods. It also missed a comparison of the efficiency of the results.

It is recommended to expand the work by describing the scientific solution applied by the authors. Describe the scientific, not technological, part. Compare the scientific results of the article with the analogous solutions of other authors. Since there is no clear scientific method, it is necessary to determine how the Bittoret protocol will be reflected in the attack and whether there will be a different effect using another TCP or UDP protocol. It is a good idea to use virtual infrastructure. The outcome is a technological solution, but it lacks scientific value. The result is not comparable with other existing scientific methods. It also missed a comparison of the efficiency of the results.

It is suggested to add to the work by explaining the scientific method the authors used. Describe the scientific, not technological, part. Compare the scientific results of the article with the analogous solutions of other authors. Since there is no clear scientific method, it is important to figure out how the Bittoret protocol will affect the attack and if another TCP or UDP protocol will have a different effect.

Reviewer 3 Report

Torrent poisoning protection with reverse proxy server

MDPI Electronics 206404

This paper presents an interesting 'after action' report on a DDOS incident that the authors experienced, and the triage undertaken. While the work done in reacting appears to have been successful, ideally I would like to see more quantification of the results, in a test environment, what do the changes lead to in terms of volume management. How frequently were requests being made by clients.  Is there an analysis on the ASN's / geolocation of clients?

I believe that the changes while ideally suited on a HA Reverse proxy, could similarly be implemented on a webserver.  Similarly, the final sentences allude to the need for a smaller error response page, but this is not quantified. This should be possible in reviewing logs, and determining a calculation of the number of requests, bytes etc.  While the SNI based solution was effective, it would be good to have some consideration presented as to how this could e be integrated with other security platforms, such as the Firewall, routing, and even appropriate configuration on the web servers.

Given this I’m left a little unclear as to the core focus of the paper. Is it a description of the incident management, and evaluation of the incident, or the suggestion of a viable technique for dealing with this class of attack ( and by extension similarly HTTP/HTTPS level attacks) ?

With this in mind,  I am left wondering is if this an appropriate Journal to have targeted the work for, rather than potentially some more specialist security/operations focussed platforms.

The paper as it stands needs some work before it can be considered for publication.  This includes checking references, and some grammatical proofreading for clarity. Further notes are contained below.

Abstract

DDOS is effective, and disruptive. This can also be done using Torrent poisoning, it is however not the only means as the current abstract implies. this should be re-worded.

Why is web being introduced? Web implies 80,443/tcp  You should define the scope f the paper clearly that the work considers the impact of DDoS on web systems ( somewhat clarified in the final sentence). Could this possibly be network flooding? The abstract needs to be clarified. lines 2-5 comprise a very long sentence, that is awkward to read. break this into two parts and this will improve the clarity.

Keywords

Is Linux needed? it is very generic, as is open source software

Capitalise Index poisoning - align with other terms

Section 1

Typically, the structure of the paper is laid out in the Introduction. This is very sparse and does not adequately define the problem space that is being addressed.

Citation for final sentence? Add dates when this was prevalent.

Section 2

line 25 - consider formatting .torrent as italics or in a fixed width font, to make it stand out?

line 27 - This -> These

line 28 - trackers -> tracking

line 29 - s/users/clients this is more accurate, as a user could have multiple clients

line 30 - Avoid providing an offhand reference to a Figure such as is done here. Be explicit as to what the reader should take away from the figure. If i there is nothing specific that the figure adds, it should be removed. This figure does not relate to the sentence its referred to in.

line 33 - consider: "The clients with that list are able to connect directly to each other using Peer-to-Peer (P2P)."

Consider moving the discussion relating to magnet links to the end once you have described the torrent file and use, and then explain that using the hash the lookup can be done. this needs to be explained, however, and linked to the threat/direct relevance to the paper.

Section 3

line 37 - consider: "A server or cluster of related servers is flooded with a large volume requests" technically they can be counted.

You need to differentiate between resource exhaustion attacks consuming processing/memory ( eg syn floods or TLS negotiations) and volumetric attacks which choke the network. What web flooding attack is being considered - there are a number which are not clarified.

line 40 - the source identification difficulty is very much dependant on the type of attack executed. Where there are spoofable datagrams, it is near impossible (with reflected/amp attacks possibly being easier), but if a full 3-way handshake happens it is possible to attribute and filter.

Section 3.1

Consider a better reference to use to define flooding attacks such as:

Zargar, S.T., Joshi, J. and Tipper, D., 2013. A survey of defense mechanisms against distributed denial of service (DDoS) flooding attacks. IEEE communications surveys & tutorials, 15(4), pp.2046-2069.

The discussion here looks at typical volumetric attacks but does not link to the web issues raised previously.

Section 4

line 64 - torrents were not rendered unusable, there was however an impact. State the date range when this activity occurred. This was a very long time ago. Are there more recent works/incidents. Consider the inclusion of how this approach has also been used for the proliferation of malware, often 'squatting' on popular software/pirate content.

line 69 - claims of the attack being devastation are over hyped. Provide evidence/explanation

Section 5

lines 71-76 - worth stating rather that a a full 3-way handshake is established, which would be clearer. the fact that a full connection is make also means the 'far side' is known to the victim server.  You should include some discussion of ports here.

lines 76-79 - very long sentence, break this down. The  retry is also not constant, with almost all clients having a backoff and periodic polling of trackers.

This needs to be explained carefully:

"These changed torrent file by itself may not render the attack successfully, since some of the 80

websites where the .torrent file are publish and shared, require a number of seeders and 81

lechers (users sharing the full files content of the torrent)."

the statement relating to full files applies to the seeders only NOT leachers (who is everyone else)

line 83 - how is this deception done, and highlight the relevance to the overall attack strategy

Section 6 - suggest retitling: "Case study - Attack of a higher education institution"

line 87 - "The institution in the case study"

                - Students and staff should suffice

lines 88-96 - proof for readability and grammar.

is the course and department relevant?

line 106 thought -> through, or just "Normal daily traffic is usually around 30 to 50 Mbps inbound and outbound"

line 111 - tree -> three

Be clear the HA was looking at heartbeat failover, rather than failure to provide service.

Figure 6  is a little small. Are you happy disclosing the client IP addresses that made the requests ?

line 144 - This is written in the tone of an incident report, with the incident under way,, but not finalised. Be specific with statement : “Hopefully there’s a solution for this issue in the TLS protocol".  You present SNI in the section following so this sentence is moot , and should be reworded or removed.

lines 173/174 - consider using a different font to indicate the ssl_fc...and hdr elements. Or just state that it has mechanisms for doing the validation.

line 198 - tcp -> TCP ; In this cases HAProxy will do a tcp request reject <-- Will generate a TCP Reset ?

Avoid splitting listings over pages, try remove unnecessary whitespace in these, the can be compacted and improve readability. Refer directly to the listing, as to what the reader should find important/relevant and ideally guide the interaction, so the reader takes the same understanding as you.

This listing is not referred to directly in the text. the relevant parts here should be explained, and possibly changes from the default bolded. Consider adding a flow chart that shows the HA proxy filtering/flow process.

Section 8

You need to clearly state /demonstrate why there is the regression back to describing the access stats form 2020. Its already been established that there is a significant spike in traffic. but user counts are not mentioned to date.

Figures 12 /13/14 I would suggest annotating these to clearly show what is important and that the readers should pay attention to.

line 228 - begging -> beginning;

line 229 - 6Th -> 6th

line 254 - as above

This section while interesting in terms of post incident analysis is questionable form a research sense. The evidence provided shows and frames the background to the system

Abbreviations

This is not something I have seen commonly, but I defer to house style for formatting.

Section 9

There are other more crude techniques, for example filtering incoming IP traffic to impacted systems  that is outside of known ASN or geolocated ranges

Section 10

Further active defences could be activated, in term of temporary firewall filters put in place for offending addresses (similar to the fail2ban concept). .  Linking a commonly used system such as fail2ban, could automate blocking of offending hosts on the n’th interaction. This reduces the load to a series of incoming SYN packets which is substantially lower, and drops the resourcing requirements.

The point made about the need for a 'small and self-contained' 404 response are valid. It would be better to demonstrate/quantify what this would  result in in terms of traffic volume and number of distinct requests. A custom response (*403?) could also be defined for the specific URL ?

The cost here is not only in bandwidth, but the TLS overhead, meaning its potentially bandwidth, and CPU based attack.

References

These are well formatted and consistent. There is some inconsistency in how DOI's are represented. For example, consider [1,3] and [6,9].  A consistent style must be used.

[5] note that this article is currently "under concern" with SAGE - https://journals.sagepub.com/doi/full/10.1177/00207209211064046

This paper does also not really address the points its being cited to support.

[8] - spell out SRUTI --> " 3rd Workshop on Steps to Reducing Unwanted Traffic on the Internet (SRUTI '07)"

[10] an RFC as a well-established documentation source, does not need an accessed date. Consider as @Techreport rather than as a web resource

[16] Check consistency wrt expanding journal names.

Round 2

Reviewer 1 Report

I see that the paper has been improved and has a better and clearer presentation than the first version. However, I have some points to be addressed:

The abstract mainly summarizes the paper's content to give readers a few seconds of insight and vision about the main contribution of that work. Therefore, usually, citing any reference is not a good idea for the abstract section. Also, in the introduction section, it would be helpful to add a couple of sentences before "This document is organized as follows......" to represent how your contribution will address the discussed security gaps. In addition, there is still no clear future vision regarding the paper's work. I found confusion in the understanding of Section 5 (HAProxy); and why that section is described after the case study? This is the tool that the authors used to improve the performance and reliability of a server environment in their case study. I see it would make sense to high light that and move it before/within Section 4.

Reviewer 2 Report

A gap that was reported in the comments was fixed. The work has been updated and supplemented. The paper with the experiment is ready for publication.

Reviewer 3 Report

Changes appear largley addressed,a dn the paper reads much better.

Have a careful look at figures 7 & 88 relating to the placement of the hilighitng boxes. For Fig 7 consider using boxes of different colours rather than tha A/B which isare poorly placed. For figure 8  I would sugegst limiting the box to the size and dates of the files of interes,  currenlty it is poorly places and covers part of the file name.
